# Efficacy of the Sentinox Spray in Reducing Viral Load in Mild COVID-19 and Its Virucidal Activity against Other Respiratory Viruses: Results of a Randomized Controlled Trial and an In Vitro Study

**DOI:** 10.3390/v14051033

**Published:** 2022-05-12

**Authors:** Donatella Panatto, Andrea Orsi, Bianca Bruzzone, Valentina Ricucci, Guido Fedele, Giorgio Reiner, Nadia Giarratana, Alexander Domnich, Giancarlo Icardi

**Affiliations:** 1Department of Health Sciences (DISSAL), University of Genoa, 16132 Genoa, Italy; panatto@unige.it (D.P.); andrea.orsi@unige.it (A.O.); icardi@unige.it (G.I.); 2Hygiene Unit, San Martino Policlinico Hospital-IRCCS for Oncology and Neurosciences, 16132 Genoa, Italy; bianca.bruzzone@hsanmartino.it (B.B.); valentina.ricucci@hsanmartino.it (V.R.); 3NG Scientific Consulting, 20091 Bresso, Italy; guido195263@gmail.com; 4APR Applied Pharma Research SA, via Corti 5, CH-6828 Balerna, Switzerland; giorgio.reiner@apr.ch (G.R.); nadia.giarratana@apr.ch (N.G.)

**Keywords:** hypochlorous acid, Sentinox, COVID-19, SARS-CoV-2, respiratory viruses, viral load, randomized controlled trial, efficacy

## Abstract

Sentinox (STX) is an acid-oxidizing solution containing hypochlorous acid in spray whose virucidal activity against SARS-CoV-2 has been demonstrated. In this paper, results of a randomized controlled trial (RCT) on the efficacy of STX in reducing viral load in mild COVID-19 patients (NCT04909996) and a complementary in vitro study on its activity against different respiratory viruses are reported. In the RCT, 57 patients were randomized (1:1:1) to receive STX three (STX-3) or five (STX-5) times/day plus standard therapy or standard therapy only (controls). Compared with controls, the log_10_ load reduction in groups STX-3 and STX-5 was 1.02 (*p* = 0.14) and 0.18 (*p* = 0.80), respectively. These results were likely driven by outliers with extreme baseline viral loads. When considering subjects with baseline cycle threshold values of 20–30, STX-3 showed a significant (*p* = 0.016) 2.01 log_10_ reduction. The proportion of subjects that turned negative by the end of treatment (day 5) was significantly higher in the STX-3 group than in controls, suggesting a shorter virus clearance time. STX was safe and well-tolerated. In the in vitro study, ≥99.9% reduction in titers against common respiratory viruses was observed. STX is a safe device with large virucidal spectrum and may reduce viral loads in mild COVID-19 patients.

## 1. Introduction

The ongoing COVID-19 pandemic has caused an unprecedented burden on healthcare systems: as of 1 May 2022, more than 500 million confirmed cases and over 6,000,000 deaths have been reported worldwide [1]. Vaccination, social distancing, and good respiratory and hand hygiene are the key individual preventive measures essential to tackle the ongoing COVID-19 pandemic [2]. Evolution of SARS-CoV-2 with the emergence of novel variants of concern (VOCs) with increased transmissibility [3] and the inability to generate sterilizing immunity to SARS-CoV-2 in most vaccinated or recovered individuals warrants the availability of effective and safe antiviral agents, especially in the community setting [4]. Moreover, other typically seasonal respiratory viruses like influenza or respiratory syncytial (RSV) viruses continue circulating (although to a lesser extent thanks to some non-specific interventions such as mask wearing and lockdowns) and co-infections have been increasingly reported [5,6], and these may be associated with poorer clinical outcomes [7].

Compared with systemic drug administration, topical delivery routes have several advantages including increased product bioavailability, reduction of side and off-target effects and improved patient convenience, which leads to increased compliance [8]. The nasal route of administration of both therapeutic and prophylactic agents against SARS-CoV-2 has been gaining particular attention [9]. Indeed, the nasal lining serves as a primary defense against inhaled pathogens, which are mainly transmitted through respiratory droplets. Virucidal activity of topical agents is further enhanced by the physical action of irrigations, since nasal rinses disrupt the viscous surface layer with associated virion particles and increase hydration of the deeper aqueous layer, therefore improving mucociliary function and mucostasis [10].

The SARS-CoV-2 viral load in the nasopharynx and nasal cavity is usually higher than in the oropharynx or the saliva [11,12]. It has been suggested [13,14] that owing to a relatively high expression of the angiotensin-converting enzyme 2 (ACE2) receptors, to which SARS-CoV-2 binds, the nasal cavity is a dominant and fertile site for early SARS-CoV-2 infection. A high nasopharyngeal (NP) viral load may contribute to the secondary transmission of SARS-CoV-2. Moreover, viral loads may be also higher among cases able to transmit to others as compared to those who do not transmit [15]. The increased decolonization of the nasal cavity may therefore reduce viral shedding and the transmissibility of SARS-CoV-2.

As shown in a recent Cochrane review on the benefits and harms of nasal sprays and mouthwashes [16], several randomized controlled trials (RCTs) with different antimicrobial solutions (e.g., cetylpyridinium chloride, chlorhexidine, chlorine dioxide, essential oils, hydrogen peroxide, hypertonic saline, nitric oxide releasing solution, povidone iodine) are ongoing. First, available data have suggested mixed and inconclusive findings. For instance, compared with the placebo saline group, patients treated with a nitric oxide nasal spray showed a 1.21 log_10_ reduction (*p* ≤ 0.02) in SARS-CoV-2 load on days 2 and 4, while this decrease (0.98 log_10_) was not significant (*p* = 0.069) on day 6 [17]. Mouth washes and gargles followed by a nasal pulverization with povidone-iodine 1% reduced SARS-CoV-2 concentration from baseline to day one in a small French RCT [18]. On the other hand, Zarabanda et al. [19] have not found any meaningful difference in viral loads of patients treated with placebo saline and 0.5% and 2% povidone-iodine nasal sprays.

Hypochlorous acid (HClO) is a powerful oxidizing antiviral, antibacterial and antifungal agent that selectively binds with the unsaturated lipid layer, disrupting the integrity of pathogens. Its virucidal effect is driven by chlorination, resulting in the formation of chloramines and nitrogen-centered radicals that damage viral nucleic acids [20,21]. The anti-SARS-CoV-2 activity of HClO has recently been demonstrated in an in vitro model [22]. HClO is naturally produced by the human immune system, has a long and well-established safety record, and is currently recommended as a disinfectant [20].

Sentinox (STX; APR, a subsidiary of RELIEF THERAPEUTICS Holding SA, Genève, Switzerland) is an acid-oxidizing solution containing HClO in a spray formulation. The antimicrobial effect of STX is driven by its unique chemico-physical characteristics: the combination of low pH and high oxidation-reduction potential reinforces the antimicrobial activity of HClO. In particular, the low pH inhibits microbial growth, while the high redox potential destabilizes membrane potential of the microorganisms and facilitates their killing [23,24]. A pre-clinical study of the solution showed >99.8% virucidal activity against SARS-CoV-2 cultivated in Vero cells in <1 min of contact time. The solution was also non-irritating [25]. However, no RCTs on the efficacy of this class III medical device in reducing SARS-CoV-2-related outcomes have been conducted so far. In this paper, the results of two studies are reported and discussed. The first study was a post-marketing RCT aimed at evaluating efficacy of the STX spray in decreasing the viral load in patients with mild COVID-19 disease and to assess its safety, tolerability and patient satisfaction. In the second in vitro study, we evaluated the virucidal activity of the STX spray against a variety of respiratory pathogens including human influenza viruses A and B, RSV, rhinovirus, adenovirus, parainfluenza virus and seasonal coronavirus.

## 2. Materials and Methods

### 2.1. Randomized Controlled Efficacy Trial

#### 2.1.1. Study Design

This prospective, randomized (1:1:1), controlled, open-label, parallel-group, single-center, phase IV study was conducted in Genoa (Northwestern Italy) between May and November 2021. The study was registered with clinicaltrials.gov (NCT04909996) (accessed on 15 April 2022) [26]. The study protocol was amended with the following main changes: (i) the extension of the enrollment period to an additional two months (driven by a sharp decrease in new SARS-CoV-2 infections); (ii) the possibility of enrolling vaccinated subjects (driven by the fact that most residents in Genoa received at least one COVID-19 vaccine dose). The study was conducted in accordance with the principles of the Declaration of Helsinki and Good Clinical and Laboratory Practices guidelines. The CONSORT (consolidated standards of reporting trials) statement [27] was adopted for the reporting. The study protocol and subsequent amendments were approved by the Ethics Committee of the Liguria Region (Genoa, Italy) (resolution 205/2021 of 20 April 2021). Prior to being enrolled (day 0), all subjects provided written informed consent. Participants could withdraw their informed consent at any time. Investigators could also withdraw subjects from the trial for medical reasons (e.g., progression to severe disease or hospitalization for any cause) at any time.

#### 2.1.2. Study Population

Potentially eligible participants were prospectively identified from a list of requests for molecular diagnostics of SARS-CoV-2, which was performed at the regional reference laboratory for COVID-19 diagnostics of the San Martino Policlinico Hospital (Genoa, Italy). In particular, community-dwelling subjects aged 18–64 years, residing in the municipality of Genoa, positive for SARS-CoV-2 in reverse transcription real-time polymerase chain reaction (RT-PCR) with cycle threshold (Ct) values ≤ 30 for at least two gene targets (see below), and not having previous positive results on rapid antigen-detecting or nucleic acid amplification tests (NAATs) were potentially eligible. Potentially eligible subjects were first pre-screened by the research staff by telephone, and those willing to participate in the study were visited (day 0) at their home by an expert physician. To be enrolled in the study, subjects had to present with a mild disease and with at least one COVID-19-attributable symptom (e.g., fever, cough, runny nose, dysosmia) arising < three days before the day of potential enrollment. Mild disease was defined according to World Health Organization (WHO) criteria [28] as symptomatic patients meeting the case definition of COVID-19 without evidence of viral pneumonia or hypoxia. The principal exclusion criteria were the following: treatment with medications with known or presumptive antiviral activity, the presence of any important co-morbidities, known hypersensitivity to any STX ingredients and its metabolites, cognitive impairment, use of a high-flow nasal cannula or non-invasive ventilation, alcohol or substance abuse, known pregnancy, breastfeeding, or ongoing or prior participation in other clinical trials. The full list of inclusion and exclusion criteria is available elsewhere [26].

#### 2.1.3. Investigational Product

STX is an acid-oxidizing sprayable formulation of HClO at 0.005% (50 ppm) in a liquid carrier solution obtained through the proprietary electrolysis process of Tehclo Technology (APR, a Relief Therapeutics Holding AG Company, Balerna, Switzerland). The solution has a low pH of 2.5–3.0, high oxidation-reduction potential (1000–1200 mV) and an HClO content > 95% of the free chlorine species. Several pre-clinical evaluations (data on file) showed that the device is non-irritating to human skin or mucosa, non-sensitizing, non-mutagenic, non-pyrogenic, not toxic up to 50 mL/kg i.e., non-phototoxic, had no keratolytic effect, and satisfied the requirements of the intramuscular implantation test. Special handling precautions were not required.

Currently, the STX nasal spray is certified in Europe as a Class III Medical Device (Certificate Nr. EPT 0477.MDD.21/4200.2) and indicated for irrigation, cleansing and moistening of the nasal cavities for (i), reducing the risk of infections caused by bacteria and viruses, including SARS-CoV-2, by lowering the nasal microbial load; (ii) symptomatic nasal care and (iii) nasal care in case of minor lesions/alterations of the nasal mucosa.

STX was delivered in a 50 mL bottle. Each dose consisted of the application of 0.5 mL of STX into each nostril according to the following instructions: (i) tilt the head slightly backwards; (ii) close one nostril and gently insert the nozzle into the other; (iii) gently squeeze the pump five times as you breathe in; (iv) switch to the other nostril and repeat; (v) wait at least 1 min and then blow the nose.

#### 2.1.4. Study Procedures

Randomization. Participants were randomized (simple randomization) on a 1:1:1 ratio in the following arms: STX-3 (administration of STX three times a day as addon to the standard physician prescribed symptomatic therapy), STX-five (administration of STX five times a day as an addon to the standard therapy) and C (control group, standard therapy only). All randomized patients received treatment according to which group they were allocated to. Owing to the study design and procedures, the allocated treatment could not be blinded to both patients and investigators.

Patient visits. Eligible patients were visited at their home. Principal study procedures by daily visit are reported in Appendix A. Briefly, on day 0, following the application of inclusion and exclusion criteria and the granting of informed consent, an expert physician collected relevant medical history data and medications being used and any COVID-19-related symptoms, performed a physical examination, and recorded principal vital signs (axillary body temperature, oxygen saturation and heart rate measured by a fingertip pulse oximeter). Subjects in all study arms were given a diary and instructions on how to fill in it. The diary consisted of an ad hoc questionnaire on symptoms (including nasal congestion, fever, dry cough, wet cough, difficulty breathing/shortness of breath, loss of taste, loss of smell, tiredness, muscle soreness/ache, sore throat, diarrhea, vomiting, conjunctivitis, headache, skin rash, discoloration of toes, chest pain), adverse events (AEs) and concomitant medications. Diaries for the intervention arms STX-3 and STX-5 had also illustrative instructions on the self-administration of the investigational product. The STX-3 and STX-5 groups were dispensed with the investigational product and provided detailed instructions on the mode of use according to the intervention group. On the following days, there was a total of three visits on days one and two (8:40 am ± 10 min, 2:40 pm ± 10 min and 8:40 pm ± 10 min), while visits on days three to six, 10 and 21 visits were made at 8:40 a.m. ± 10 min only. AEs, concomitant medications and vital signs were assessed at all visits.

To ensure the treatment compliance, on each treatment day (1–5), the research staff telephoned patients of groups STX-3 and STX-5 10 min before each planned spray administration.

Starting from day one, all participants were taken an NP swab in both nostrils. On days one and two, three daily swabs were performed at 8:40 a.m. ± 10 min, 2:40 p.m. ± 10 min and 8:40 p.m. ± 10 min. On subsequent days, only one NP swab was performed at 8:40 a.m. ± 10 min. Before swabbing, subjects were asked to blow the nose. In order to harmonize the swabbing procedure, all research staff members were provided unique instructions and an educational video [29]. Each cotton flock was eluted in the universal transport medium (UTM, Copan Italia S.p.A., Brescia, Italy) and transported to the laboratory.

Following completion of the treatment (days one to five), a total of three follow-up visits on days six, 10 and 21 were performed. At each follow-up visit, both clinical examination and NP swabbing were performed. Moreover, on day six the patient diary was assessed for completeness and was retrieved together with the rest of the investigational product. On the same day and following a physical examination, patients in groups STX-3 and STX-5 were asked to rate the overall tolerability of the investigational product and their satisfaction. This was performed by means of a 10-cm visual analogue scale (VAS) and 5-point (very unsatisfied, unsatisfied, neither unsatisfied nor satisfied, satisfied, very satisfied) Likert scale, respectively.

Reverse transcription real-time polymerase chain reaction (RT-PCR). Each specimen collected was tested by means of both qualitative and quantitative RT-PCR at the regional reference laboratory for COVID-19 diagnostics of the San Martino Policlinico Hospital (Genoa, Italy). First, total RNA was extracted by means of the STARMag Universal Cartridge Kit (Seegene Inc., Seoul, Korea) on the automated Nimbus IVD (Seegene Inc., Seoul, Korea) platform according to the manufacturer’s instructions [30]. In particular, 200 µL of a fresh sample was extracted and eluted with 100 µL of elution buffer and set up for RT-PCR. Qualitative RT-PCR was then run on a CFX96 thermal cycler (Bio-Rad Laboratories, Hercules, CA, USA) with an Allplex 2019-nCoV multiplex assay (Seegene Inc., Seoul, Korea). This kit simultaneously detects different genes targeting the nucleoprotein (N), RNA-dependent RNA-polymerase (RdRp)/Spike (S) and envelope (E) regions. Briefly, the amplification step was first performed at 50 °C for 20 min, followed by 95 °C for 15 min and 45 cycles at 95 °C for 10 s, 60 °C for 15 s with first acquisition and 72 °C for 10 s with second acquisition. For each RT-PCR run, 5 µL of the extracted RNA in a final volume of 20 µL was used. According to the manufacturer, the analytical specificity of this method is 100% [30]. For the primary endpoint, samples showing Ct values < 40 for at least two gene targets were considered positive [31]. However, considering that samples with high Ct values are usually not recoverable in cell culture and therefore not infectious [32], we also used the positivity cut-off of 35 (see also below) [33,34].

Quantitative RT-PCR was then performed in order to quantify the viral load (copies/mL) by using the Quanty COVID-19v2 assay (Clonit Srl, Milan, Italy) according to the manufacturer’s instructions [35]. Briefly, this assay simultaneously detects three N gene regions (N1, N2 and N3), and the quantitation is performed automatically by interpolation of the patient’s Ct values with the standard curve obtained following amplification of five standards containing 10^1^, 10^2^, 10^3^, 10^4^, and 10^5^ copies/mL of a synthetic viral N1-encoding RNA [36].

Independent evaluations [36,37] have reported an optimal agreement between Allplex 2019-nCoV and Quanty COVID-19v2 assays. However, in the case of inconclusive or discordant results, both qualitative and quantitative RT-PCRs were repeated.

#### 2.1.5. Study Outcomes and Definitions

The primary endpoint was the efficacy of STX in reducing viral load at any time during the days of treatment (days one to five), as determined by quantitative RT-PCR. The absolute change in viral loads was expressed as the difference between *t*_0_ and *t*_d_, where *d* is the day of follow-up. The secondary efficacy endpoint was set in order to compare the time length to negativization between the study arms. As described earlier, subjects were defined as negative according to two different Ct cut-off definitions, namely 40 [31] and 35 [33,34]. On the basis of baseline Ct values, specimens were also categorized into “high viral load” (Ct < 20) and “medium viral load” (Ct 20–30) samples. Other secondary outcomes included the frequency of AEs, the tolerability measured on VAS and patient satisfaction measured on a 5-point anchored Likert scale.

#### 2.1.6. Sample Size

The sample size was determined *a priori* on the basis of a clinically relevant reduction in viral load between groups STX-3/STX-5 and C. The clinically relevant reduction was set to 1.5 log_10_ copies/mL with a standard deviation (SD) of 1.2 log_10_ copies/mL. A sample size of 14 patients per group was necessary to achieve an 80% power and Bonferroni-corrected two-tailed *α* of 0.025 of detecting the prespecified difference in viral loads. Assuming a dropout rate of 25%, the final sample size was 57 (19 per group) patients.

#### 2.1.7. Statistical Analysis

All analyses were performed according to the intention-to-treat (ITT) principle. Per-protocol (PP) population analysis was also performed to determine the extent to which the missing data might have influenced the results. The overall effect of STX spray was also verified by pooling the two treatment arms.

Individual viral loads were transformed as log_10_ (copies/mL + 1) to account for natural zeros. Continuous variables were expressed as means ± SDs or medians with ranges. Categorical variables were expressed as proportions. A Fisher-Freeman-Halton exact test was used to test differences in proportions. The Cochran–Mantel–Haenszel test was used to compare between-group differences in negativization rates and the corresponding effect sizes were expressed as risk ratios (RRs) with 95% confidence intervals (CIs). For the primary endpoint of change in viral loads, treatment effects were estimated by applying mixed linear models, with analysis of co-variance (ANCOVA) methods with type III orthogonal sum of squares. Time point, treatment regimen and time-per-treatment interaction term were set as fixed effects, while the baseline viral load was used as a covariate. The time points were specified as repeated measures. In the matrix model, the compound symmetry for the observations within each patient was assumed.

All analyses were performed in SAS v9.4 (SAS Institute, Cary, NC, USA).

### 2.2. In Vitro Study

#### 2.2.1. Viruses

A total of seven different respiratory viruses were tested; these were obtained from the American Type Culture Collection (ATCC) (Manassas, VA, USA). These viruses were: influenza A(H1N1) (strain A/Puerto Rico/8/34, ATCC VR-1469); influenza B (strain B/Hong Kong/5/72, ATCC VR-823); RSV A (Long strain, ATCC VR-26); human rhinovirus (strain 1059, ATCC VR-284); human adenovirus type 5 (strain adenoid 75, ATCC VR-5); human parainfluenza virus type 3 (strain C243, ATCC VR-93); seasonal coronavirus (strain 229E, ATCC VR-740).

#### 2.2.2. Virucidal Assays

Working stocks of viruses were prepared by passaging in MDCK (for both influenza viruses), Hep-2 (for RSV), MRC-5 (for rhinovirus), A-549 (for adenovirus), MDBK (for parainfluenza virus) and WI-38 (for seasonal coronavirus) cell cultures. The stock virus cultures were adjusted to contain 1% fetal bovine serum (FBS). The cells were seeded into multi-well cell culture plates and maintained at 36–38 °C in a humified atmosphere of 5–7% CO_2_. On the day of testing, cell cultures were inspected for their integrity and confluency. Virus-specific test media used in the study are reported in Appendix A. The heat-inactivated FBS was used as a neutralizer.

STX was used in its commercially available formulation. For the treatment of virus suspension, a 1.80 mL aliquot of the test substance was dispensed into a separate tube and mixed with 200 µL of the virus suspension, vortexed for 10 s and incubated at 35 °C. The exposure time assayed ranged from 15 to 120 s. Immediately after each exposure time, a 100 µL aliquot was removed from each tube and tittered by 1:10 serial dilutions. In parallel, water was tested as the virus control. Neutralization and cytotoxicity controls were also tested to ensure that the virus inactivation did not continue after the pre-specified contact time.

The above-described cell lines, which exhibit a cytopathic effect in the presence of corresponding viruses, were used as the indicator cell line in the infectivity assays. Cells cultured in multi-well dishes were inoculated in quadruplicate with 100 µL of the dilutions prepared from the test and control groups. Uninfected indicator cell cultures (controls) were inoculated with test medium alone. The cultures were scored periodically for the presence or absence of cytopathic effect, cytotoxicity and viability.

Infectivity and cytotoxicity titers were expressed as −log_10_ of the median tissue culture infectious dose (TCID_50_), and computed by applying the Spearman-Karber’s method [38]. Finally, the log_10_ reduction value (LRV) of the STX vs. negative control was calculated.

## 3. Results

### 3.1. Randomized Controlled Efficacy Trial

#### 3.1.1. Baseline Characteristics of Patients

From 20 May to 9 November 2021, a total of 57 patients were enrolled. All 57 patients were randomized and allocated to groups STX-3 (*n* = 18), STX-5 (*n* = 20) and C (*n* = 19). There were three consent withdrawals (one in each group). A total of five patients (2 in group STX-5 and three in group C) terminated the participation earlier due to the worsening of clinical conditions. Therefore, the final ITT population included 54 (17, 19 and 18 in groups STX-3, STX-5 and C, respectively) patients, while the PP population included 49 patients (17, 17 and 15 in groups STX-3, STX-5 and C, respectively) (Figure 1).

Table 1 reports the principal characteristics of the ITT population. Briefly, the distribution of participants according to age, presence of underlying medical conditions, concomitant medications and symptomaticity were approximately equal. At enrollment, the most commonly used concomitant medications were non-steroidal anti-inflammatory drugs (18.5%) and antibiotics (13.0%). In the group C, four (22.2%) patients took at least one concomitant medication (antibiotics, non-steroidal and steroidal anti-inflammatory drugs were prescribed to three, two and two patients, respectively). The corresponding proportions in groups STX-3 and STX-5 were 41.2% and 52.6%, respectively. The most commonly reported COVID-19-related symptoms were dry cough (64.8%), tiredness (59.3%), nasal congestion (55.6%), myalgia (51.9%), headache (50.0%), fever (50.0%), ageusia (48.1%), wet cough (40.7%), anosmia (38.9%) and sore throat (35.2%). Although not statistically significant (*p* = 0.40), the viral load at visit 0 was higher in the STX-5 group than in the STX-3 and C groups (Table 1).

#### 3.1.2. Efficacy

As reported in Figure 2, there was a constant (*p* < 0.0001) decay in viral loads independently of the ITT study arm. The overall effect of treatment was not significantly (*p* = 0.95) different from group C. However, though non-significant, by the end of treatment on day five, patients in group STX-3 showed a 1.02 (95% CI: −0.35–2.40; *p* = 0.14) log_10_ reduction in viral load, as compared with group C. The corresponding log_10_ reduction in the STX-5 group was 0.18 (95% CI: −1.19–1.54; *p* = 0.80). The effects of viral load at screening (*p* = 0.26) and time-per-treatment interaction (*p* = 0.59) were not statistically significant. Similar results were observed in the PP population (Appendix A).

A visual inspection of the distribution of viral loads at *t*_0_ identified several influential points (outliers), and all of these had high viral loads (Ct < 20). For this reason, a stratified analysis was conducted by dividing subjects on the basis of Ct values at screening into two categories, namely medium (Ct 20–30) and high (Ct < 20) viral loads. However, it was possible to conduct the inferential analysis only for medium viral load samples, as the number high viral load samples was insufficient. As shown in Figure 3, when only including subjects with medium viral loads (*n* = 39, 13 in either group), the efficacy of STX was more pronounced in both treatment arms. In particular, on the last day of treatment (day five), compared with group C, subjects in the STX-3 group showed a significant (*p* = 0.016) 2.01 (95% CI: 0.37–3.65) log_10_ reduction in viral loads. The corresponding reduction (0.53; 95% CI: −1.13–2.18) in the STX-5 group was not statistically significant (*p* = 0.53).

For the secondary efficacy endpoint, the study groups were compared in terms of negativization by using two different cut-offs. As shown in Figure 4, the negativization was faster in both treatment arms than in group C. In particular, by the end of treatment (day five), the proportion of negativized subjects (Ct > 40) was 29%, 11% and 11% in the STX-3, STX-5 and C groups, respectively. By applying the Ct cut-off of 35, the corresponding percentages were 35%, 21% and 17%, respectively. As demonstrated by the Cochran–Mantel–Haenszel test, the proportion of negativized subjects in group STX-3 vs. group C was significantly higher by applying both >40 and >35 Ct value-based cut-offs with RRs of 1.62 (95% CI: 1.23–2.15) and 1.65 (95% CI: 1.31–2.08), respectively. Although not statistically significant, a similar trend was observed for the comparison of groups STX-5 and C [Ct > 40: RR 1.23 (95% CI: 0.91–1.66); Ct > 35: RR 1.22 (95% CI: 0.96–1.56)]. Finally, when both treatment arms were pooled, the general association proved significant independently from the Ct value cut-off considered [Ct > 40: 1.41 (95% CI: 1.08–1.85); Ct > 35: RR 1.43 (95% CI: 1.14–1.78)]. Similar results were obtained in the PP population (Appendix A).

#### 3.1.3. Safety, Tolerability and Satisfaction

During the whole follow-up period, a total of 19 AEs were recorded. Of these, only one (5.3%) AE (irritation of the nasal mucosa) was judged to be probably related to the study treatment. As shown in Table 2, the proportion of subjects with at least one AE was similar among the study groups (*p* > 0.99). The most frequent adverse events were cough and sore throat (10.5% each). A total of 4 (21.1%) AEs were classified as serious (0, 1 and 3 in groups STX-3, STX-5 and C, respectively), but were judged unrelated to the study treatment. All serious AEs were associated with a disease progression to moderate-to-severe pneumonia. Notably, the frequency of severe adverse events leading to hospitalization due to disease progression in the treatment groups was about six times lower (2.8% vs. 16.7%) than in the control group.

For what concerns the tolerability measured on a 1-to-10 VAS, the median scores were 10 in all study arms. Analogously, no between-group difference in patient satisfaction was observed: 94.1%, 100% and 100% of participants in groups STX-3, STX-5 and C were satisfied or very satisfied with treatment.

### 3.2. In Vitro Study

Cytotoxicity was not observed for all viruses tested and any dilution tested (≤1.50 log_10_). In all experiments, the neutralization control (non-virucidal level of the test substance) indicated that the test substance was neutralized at ≤1.50 log_10_. LRVs following exposure to the STX solution are reported in Table 3. In the virucidal assays performed, the STX solution demonstrated ≥ 99.9% reduction in the stock virus titers independently of both the virus tested and exposure contact time.

## 4. Discussion

To our knowledge, this is the first RCT on the efficacy of a HClO-based solution in reducing the SARS-CoV-2 viral load in patients with mild COVID-19 disease. In the primary analysis, compared with the control group, subjects treated with STX showed a 0.5–1 log_10_ reduction (approximately 90% relative reduction) in SARS-CoV-2 concentration; this decrease was not statistically significant, likely for the reasons described below. On the other hand, when influential outliers were removed, participants in group STX-3 showed a significant viral load reduction of 2.01 log_10_ copies/mL. Analogously, treatment with STX was generally associated with a shorter time to negativization, which is essential to limit the probability of spreading the virus. The STX spray was safe and well-tolerated in both treatment arms. In the complimentary in vitro study, we also demonstrated a high level of virucidal activity of the STX solution against a variety of other than SARS-CoV-2 respiratory pathogens including influenza A and B, RSV, rhinovirus, adenovirus, parainfluenza virus and seasonal coronavirus. These pathogens are responsible for an important clinical and socioeconomic burden worldwide [39].

Besides COVID-19, the efficacy of low-concentration HClO-based nasal irrigations has been evaluated in some RCTs carried out mainly in the field of otorhinolaryngology. In particular, it has been found to be effective against some clinical endpoints relative to chronic rhinosinusitis refractory to medical therapy [40] or after functional endoscopic sinus surgery [41], allergic rhinitis [42] and pediatric chronic sinusitis [43]. Notably, reduction in terms of the bacterial (e.g., *S. epidermidis*, *S. pneumoniae*, *K. pneumoniae*, *S. aureus)* culture positivity rate among adult patients treated with HClO for four weeks were substantially higher (59% vs. 33%) than in those receiving placebo; however, like in the present study and owing to a small sample size, this difference did not reach an *α* < 0.05 [40]. The present study therefore contributes to the available experimental evidence on the potential additive clinical benefits of low-concentration HClO with its well-documented broad antimicrobial spectrum and safety record [20,21,22].

Despite the observed reduction in viral loads, the primary study endpoint was not met. A subsequent exploratory analysis identified some likely drivers of this finding. Thus, we observed a very high variability in terms of initial viral loads, whose baseline levels differed by approximately five orders of magnitude. This difference reflects both within- and between-subject variations in the viral load curve [44]. On the other hand, when the analysis was restricted to subjects with “medium” viral loads (Ct 20–30), the viral load reduction observed in group STX-3 exceeded the pre-specified superiority criterion, suggesting a significant effect of the outliers. This result may be ascribed to the individual differences in viral kinetics. Indeed, following an initial exponential growth and reaching the peak of viral concentration, a decline in viral loads begins. This, however, typically follows a biphasic curve, i.e., an initial slow exponential decay and subsequent fast decline leading to viral clearance [45,46]. This biphasic decay model also corroborates the observed non-significant decrease in the STX-5 group as compared with the control group: indeed, the average baseline viral load was the highest in group STX-5. Moreover, the observed variability may be driven by the SARS-CoV-2 population circulated during the study period. For instance, the first two months of the trial were characterized by a predominance of the Alpha VOC, while from June 2021 the Delta VOC started to replace the Alpha VOC [47]. It has been shown [48] that the Delta VOC is associated with a higher viral load in both vaccinated and unvaccinated individuals.

Notably, three of 18 patients (16.7%) in the control group terminated the trial early as a result of clinical worsening that resulted in hospitalization (two patients were admitted to the intensive care unit). By contrast, considering both treatment groups, only one out of 36 (2.8%) patients reported a severe AE leading to hospitalization due to disease progression. Although the outcome of the clinical progression was not pre-specified, it deserves further investigation. It is indeed plausible that the reduction in viral loads in the upper respiratory tract may reduce and/or slow down virus propagation to the lower respiratory tract.

The main strength of this RCT is the two-step approach for the SARS-CoV-2 molecular diagnostics adopted, in which both qualitative (for virus detection) and quantitative (for viral load quantification) RT-PCR was used. This approach is highly recommended for studies focusing on viral kinetics [49]. Indeed, the relationship between the SARS-CoV-2 viral load (copies/mL) and infectiousness is still poorly understood. By contrast, as reviewed by Jefferson et al. [32], samples with high Ct values (e.g., >35) in qualitative RT-PCR assays, which are significantly more sensitive than cultural methods, are likely associated with non-viable viral particles. It has been shown [36] that the correlation between Ct values determined by Allplex 2019-nCoV and Quanty COVID-19v2 (qualitative and quantitative RT-PCR assays used also in the present study) was high but not perfect (*ρ* = −0.72–−0.92, depending on the gene target); moreover, most NP samples with Ct > 35 showed viral loads < 5 log_10_ copies/mL [36]. In a recent model [50], it has been estimated that the virus transmission with loads < 6 log_10_ copies/mL is very low. In this regard, our study demonstrated that treatment with the STX spray was associated with a shorter time to negativization, which is essential to reduce the probability of virus transmission.

The present RCT suffers from some shortcomings as well. First of all, as the study was designed in a period when little data on SARS-CoV-2 viral kinetics were available and different viral strains were circulating, the assumed statistical dispersion of individual viral loads was underestimated. Second, despite our efforts to uniform the NP swabbing procedure, some within- and between-operator heterogeneity in performance is likely. These variations may be further driven by between-patient differences in nasal anatomy, potentially swollen nasal mucosa or earlier retraction of an incompletely saturated mucus swab due to the patient’s discomfort [51]. Third, the spray was self-administered by patients under unsupervised conditions. Although each subject was constantly reminded to administer the treatment according to the protocol, we cannot completely rule out compliance bias. Indeed, the partial noncompliance in RCTs is a frequent phenomenon and may dramatically decrease statistical power [52,53]. This, therefore, may also explain both the higher than expected variability in viral loads and the absence of the dose–response association. Finally, the study was conducted during the period when Alpha and Delta VOCs circulated. However, considering a broad virucidal and bactericidal spectrum of HClO [20,21], it is highly likely that the STX activity would be similar against other VOCs.

In conclusion, owing to the high variability of baseline viral loads, the primary endpoint of this RCT was not met, the STX spray self-administered three times a day significantly reduced the viral load by about 2 log_10_ copies/mL in subjects with medium viral loads, which are typical for subjects with mild-to-moderate COVID-19 who usually require only outpatient medical attention. Following appropriate counselling performed by general practitioners (GPs), outpatients may therefore be considered the primary target for STX. Furthermore, throughout the whole follow-up period, no safety concerns emerged, confirming its current designation as an over-the-counter medical device. The in vitro study demonstrated a high level of virucidal activity of the STX spray against a variety of respiratory viruses, including influenza and RSV. These promising results support future larger-scale clinical studies in order to assess whether the STX spray is also effective in the primary prevention of both symptomatic and asymptomatic SARS-CoV-2, influenza, RSV and other acute respiratory infections in the at-risk population. These pragmatic trials may, for example, compare the incidence of clinically diagnosed influenza-like illness (with a subsequent molecular diagnostics of the causative agent) between the cohorts of treated and non-treated subjects. Analogously, future research may investigate and compare preventive and treatment effects of different pharmaceutical forms and/or administration routes (e.g., nasal vs. oral) of the low-concentration HClO-based solutions.

## Figures and Tables

**Figure 1 viruses-14-01033-f001:**
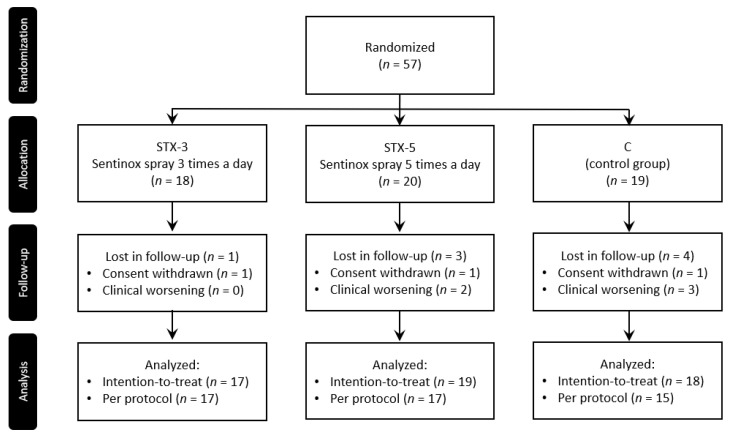
Flow chart of participants through the study.

**Figure 2 viruses-14-01033-f002:**
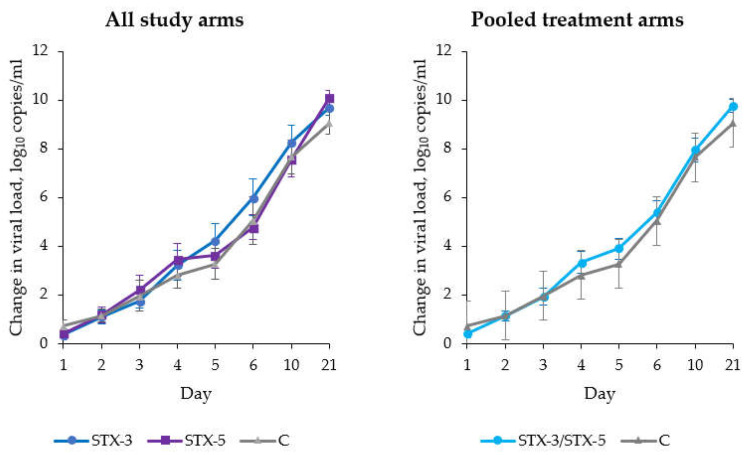
Absolute change in viral loads in the intention-to-treat (ITT) population (*n* = 54), by study arm and day of follow-up (vertical bars represent standard errors).

**Figure 3 viruses-14-01033-f003:**
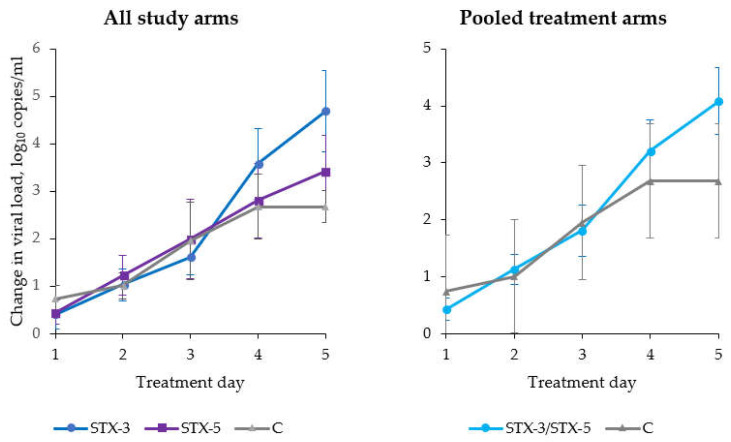
Absolute change in viral loads in intention-to-treat (ITT) population by excluding subjects with high baseline viral loads (*n* = 15), by study arm and day of treatment (vertical bars represent standard errors).

**Figure 4 viruses-14-01033-f004:**
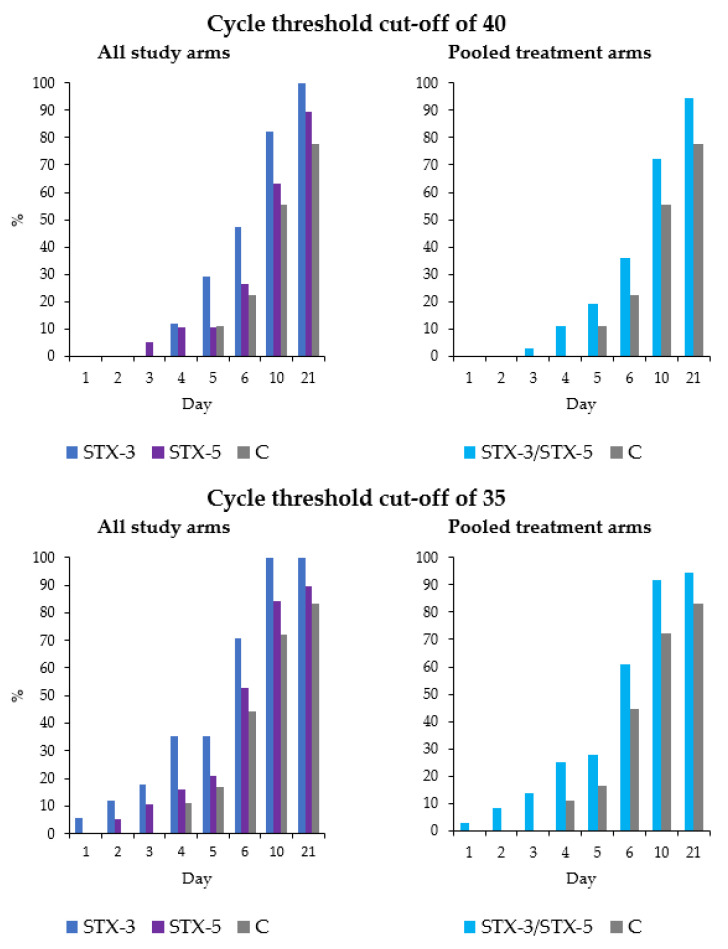
Proportion of negativized subjects in the intention-to-treat (ITT) analysis, by negativization definition, study arm and day of follow-up.

**Table 1 viruses-14-01033-t001:** Sociodemographic and clinical characteristics of patients.

Characteristic	STX-3 (*n* = 17)	STX-5 (*n* = 19)	C (*n* = 18)	Total (*n* = 54)
Age, mean ± SD (range)	37.0 ± 12.3 (21–60)	40.3 ± 14.5 (19–57)	42.7 ± 14.1 (18–63)	40.1 ± 13.7 (18–63)
Sex, % (*n*) females	23.5 (4)	52.6 (10)	27.8 (5)	35.2 (19)
Underlying medical conditions, % (*n*)	5.9 (1)	10.5 (2)	5.6 (1)	7.4 (4)
≥1 concomitant medication, % (*n*)	41.2 (7)	52.6 (10)	22.2 (4)	38.9 (21)
*n* of symptoms, median (range)	5 (1–10)	5 (1–10)	7 (3–10)	6 (1–9)
Viral load at *t*_0_, log_10_ mean ± SD (range)	9.9 ± 1.2 (8.1–11.4)	10.4 ± 1.1 (7.7–12.1)	9.9 ± 1.2 (7.1–11.6)	10.1 ± 1.2 (7.1–12.1)

**Table 2 viruses-14-01033-t002:** Frequency of adverse events during the follow-up, by study arm.

Adverse Event	STX-3 (*n* = 17)	STX-5 (*n* = 19)	C (*n* = 18)
Any, % (*n*)	17.6 (3)	21.1 (4)	22.2 (4)
Any related, % (*n*)	5.9 (1)	0 (0)	0 (0)
Any serious, % (*n*)	0 (0)	5.3 (1)	16.7 (3)

**Table 3 viruses-14-01033-t003:** Virucidal activity of Sentinox, by virus and contact time.

Virus	Contact Time, s	Log_10_ Viral Load Reduction	% Viral Reduction
Influenza virus A(H1N1)	15	≥5.75	≥99.9998
	55	≥5.75	≥99.9998
Influenza virus B	15	≥3.00	≥99.9
	55	≥3.00	≥99.9
RSV A	30	≥4.00	≥99.99
	120	≥4.00	≥99.99
Rhinovirus 14	30	≥3.25	≥99.94
	120	≥3.00	≥99.9
Adenovirus 5	15	6.00	≥99.9999
	55	≥6.25	≥99.99994
Parainfluenza virus 3	15	≥5.25	≥99.9994
	55	≥4.75	≥99.9998
Coronavirus 229E	15	≥3.00	≥99.9
	55	≥3.00	≥99.9

## Data Availability

Not applicable.

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
