# Peer review of "Efficacy of the Sentinox Spray in Reducing Viral Load in Mild COVID-19 and Its Virucidal Activity against Other Respiratory Viruses: Results of a Randomized Controlled Trial and an In Vitro Study"

_viruses, 2022, doi:10.3390/v14051033_

Round 1

Reviewer 1 Report

This study reports the results of a clinical trial of Sentinox (STX), an acid-oxidizing sprayable solution formulation of HClO 50 ppm in a liquid carrier solution obtained through a proprietary electrolysis process. The mechanism of bactericidal action is stated to be via selective binding with the unsaturated lipid layer, thus disrupting membrane integrity. The virucidal effect is claimed to be driven by chlorination resulting in the formation of chloramines and nitrogen-centered radicals that damage viral nucleic acids. STX was administered as a 3 dose or 5 dose regimen via nasal spray. The study involved subjects aged 18–64 years (n=54 total) residing in the municipality of Genoa that tested positive for SARS-CoV-2 by RT-PCR with Ct≤30 for at least two gene targets and not having previous positive rapid antigen test or nucleic acid amplification tests. The proportion of subjects turned negative by the end of treatment (day 5) was significantly higher in STX-3 compared to controls, suggesting a shorter virus clearance time. STX was also reported to be safe and well-tolerated. There was also an in vitro study performed involving several respiratory viral pathogens and mammalian cell lines involving exposing the pathogens to STX. The authors report significant decreases in virus titer.

Overall this is a well written manuscript. My comments are as follows.

The authors should include a description of the standard therapies that were administered to the control group.

The in vitro work is not sufficiently described. For example, how the cells were used in these experiments is not explained. These experiments need to be better explained.

There is no mention of STX toxicity in vitro. This is an important consideration if the data on virucidal activity is to be interpreted.

The authors state that ’the STX spray self-administered 3 times a day significantly reduced the viral load by about 2 log10 copies/ml in subjects with medium viral loads, which are typical for subjects with mild-to-moderate COVID-19.’ How would such patients be targeted? The authors should expound on how they see this drug could be applied clinically, especially since they state that further trials are warranted. Alternatively, they should expand on how these future trials may look like.

The authors further state that these data ‘support future larger-scale clinical studies in order to assess whether the STX spray is also effective in primary prevention of SARS-CoV-2, influenza, RSV and other acute respiratory infections.’ The meaning of ‘primary prevention’ should be clarified.

They also say that the ‘primary endpoint of this RCT was not met.’ The RCTs of this trial should be listed and discussed.

It is not clear why the in vitro study was included. Was the efficacy of these compounds not assessed in vitroprior to use in humans? Further, this is primarily a clinical paper. If STX is not efficacious in humans, the data in vitro is not very interesting unless there are mechanistic insights to be had.

Author Response

Comment: This study reports the results of a clinical trial of Sentinox (STX), an acid-oxidizing sprayable solution formulation of HClO 50 ppm in a liquid carrier solution obtained through a proprietary electrolysis process. The mechanism of bactericidal action is stated to be via selective binding with the unsaturated lipid layer, thus disrupting membrane integrity. The virucidal effect is claimed to be driven by chlorination resulting in the formation of chloramines and nitrogen-centered radicals that damage viral nucleic acids. STX was administered as a 3 dose or 5 dose regimen via nasal spray. The study involved subjects aged 18–64 years (n=54 total) residing in the municipality of Genoa that tested positive for SARS-CoV-2 by RT-PCR with Ct≤30 for at least two gene targets and not having previous positive rapid antigen test or nucleic acid amplification tests. The proportion of subjects turned negative by the end of treatment (day 5) was significantly higher in STX-3 compared to controls, suggesting a shorter virus clearance time. STX was also reported to be safe and well-tolerated. There was also an in vitro study performed involving several respiratory viral pathogens and mammalian cell lines involving exposing the pathogens to STX. The authors report significant decreases in virus titer.

Overall this is a well written manuscript. My comments are as follows.

Reply: Thank you for your interest in our paper. All your comments have been addressed.

Comment: The authors should include a description of the standard therapies that were administered to the control group.

Reply: This description has been now provided.

Comment: The in vitro work is not sufficiently described. For example, how the cells were used in these experiments is not explained. These experiments need to be better explained.

Reply: As required, methods for the in vitro study have been described more in detail.

Comment: There is no mention of STX toxicity in vitro. This is an important consideration if the data on virucidal activity is to be interpreted.

Reply: Thank you for this comment. Results on the STX cytotoxicity have been now reported.

Comment: The authors state that ’the STX spray self-administered 3 times a day significantly reduced the viral load by about 2 log10 copies/ml in subjects with medium viral loads, which are typical for subjects with mild-to-moderate COVID-19.’ How would such patients be targeted? The authors should expound on how they see this drug could be applied clinically, especially since they state that further trials are warranted. Alternatively, they should expand on how these future trials may look like.

Reply: As suggested, we have now described how some patients may be targeted, including their clinical designation. We have also provided an example of a future clinical trial.

Comment: The authors further state that these data ‘support future larger-scale clinical studies in order to assess whether the STX spray is also effective in primary prevention of SARS-CoV-2, influenza, RSV and other acute respiratory infections.’ The meaning of ‘primary prevention’ should be clarified.

Reply: The term primary prevention means prevention of an infection in at-risk populations; this has been now explicated.

Comment: They also say that the ‘primary endpoint of this RCT was not met.’ The RCTs of this trial should be listed and discussed.

Reply: The trial registration number was provided in both the abstract and methods.

Comment: It is not clear why the in vitro study was included. Was the efficacy of these compounds not assessed in vitro prior to use in humans? Further, this is primarily a clinical paper. If STX is not efficacious in humans, the data in vitro is not very interesting unless there are mechanistic insights to be had.

Reply: Contrary to the in vitro virucidal activity of STX against SARS-CoV-2 (see Giarratana et al. [25] discussed in the paper), it is unknown whether STX is efficacious against other common respiratory viruses. Indeed, following their relative disappearance from the epidemiological scene, the incidence of viruses such as influenza and RSV has been recently increased. Taken together, we believe that inclusion of the in vitro study would further benefit the readers of Viruses.

Reviewer 2 Report

The papers report the randomized controlled trial (RCT) on the efficacy of Sentinox (STX) in reducing viral load in mild COVID-19 patients and a complimentary in vitro study on its activity against different respiratory viruses. The overall presentation is good. However, I have some suggestions for the authors.

  1. The authors must pay attention to English language (Article usage, pronoun usage, spelling, grammar, spacing between the words, formatting, etc.)
  2. A short introduction about COVID-19, its stats, and research gaps must be provided in the introduction part.
  3. A highlight of the possible effects of Sentinox (STX) against new variants of SARS-COV-2 is recommended in the discussion part.
  4. A short discussion about the possibility of preparing different dosage forms (mouth wash, oral spray, liquid, etc.) may be useful to the readers to design further studies on the subject matter.
  5. The author must provide some recommendations based on the outcomes of the study.  

Author Response

Comment: The papers report the randomized controlled trial (RCT) on the efficacy of Sentinox (STX) in reducing viral load in mild COVID-19 patients and a complimentary in vitro study on its activity against different respiratory viruses. The overall presentation is good. However, I have some suggestions for the authors.

Reply: Thank you for your interest in our paper. All your comments have been addressed.

Comment: The authors must pay attention to English language (Article usage, pronoun usage, spelling, grammar, spacing between the words, formatting, etc.)

Reply: The whole manuscript has been now reviewed.

Comment: A short introduction about COVID-19, its stats, and research gaps must be provided in the introduction part.

Reply: As suggested, a brief introduction to the COVID-19 pandemic has been provided early in the text.

Comment: A highlight of the possible effects of Sentinox (STX) against new variants of SARS-COV-2 is recommended in the discussion part.

Reply: We have now highlighted possible effects of Sentinox against other variants of SARS-CoV-2.

Comment: A short discussion about the possibility of preparing different dosage forms (mouth wash, oral spray, liquid, etc.) may be useful to the readers to design further studies on the subject matter.

Reply: We have now provided suggestions for future study designs that would include different treatment formulations.

Comment: The author must provide some recommendations based on the outcomes of the study. 

Reply: As required, we have now added the primary STX population target.